# Homocysteine, Cognitive Functions, and Degenerative Dementias: State of the Art

**DOI:** 10.3390/biomedicines10112741

**Published:** 2022-10-28

**Authors:** Simona Luzzi, Veronica Cherubini, Lorenzo Falsetti, Giovanna Viticchi, Mauro Silvestrini, Alessio Toraldo

**Affiliations:** 1Neurology Unit, Department of Experimental and Clinical Medicine, Polytechnic University of Marche, 60126 Ancona, Italy; 2Internal and Subintensive Medicine Department, Azienda Ospedaliero-Universitaria “Ospedali Riuniti” di Ancona, 60126 Ancona, Italy; 3Neurology Unit, Azienda Ospedaliero-Universitaria “Ospedali Riuniti” di Ancona, 60126 Ancona, Italy; 4Department of Brain and Behavioral Sciences, University of Pavia, 27100 Pavia, Italy; 5Milan Center for Neuroscience (NeuroMI), 20126 Milan, Italy

**Keywords:** homocysteine, hyperhomocysteinemia, cognitive functions, dementia, degenerative dementias, Alzheimer’s disease, non-Alzheimer dementias

## Abstract

There is strong evidence that homocysteine is a risk factor not only for cerebrovascular diseases but also for degenerative dementias. A recent consensus statement renewed the importance and the role of high levels of homocysteine in cognitive decline in several forms of degenerative dementia, such as Alzheimer’s disease. Although the molecular mechanisms by which homocysteine causes cell dysfunction are known, both the impact of homocysteine on specific cognitive functions and the relationship between homocysteine level and non-Alzheimer dementias have been poorly investigated. Most of the studies addressing the impact of hyperhomocysteinemia on dementias have not examined the profile of performance across different cognitive domains, and have only relied on screening tests, which provide a very general and coarse-grained picture of the cognitive status of the patients. Yet, trying to understand whether hyperhomocysteinemia is associated with the impairment of specific cognitive functions would be crucial, as it would be, in parallel, learning whether some brain circuits are particularly susceptible to the damage caused by hyperhomocysteinemia. These steps would allow one to (i) understand the actual role of homocysteine in the pathogenesis of cognitive decline and (ii) improve the diagnostic accuracy, differential diagnosis and prognostic implications. This review is aimed at exploring and revising the state of the art of these two strictly related domains. Suggestions for future research are provided.

## 1. Introduction

Degenerative dementias are a group of neurological diseases with unknown cause and characterized by progressive cognitive impairment with loss of functional autonomy [1]. Their incidence and prevalence are destined to increase in the near future [2,3]. Currently available drugs can only help to slow down cognitive decline [4,5]. Despite research efforts, disease-modifying therapies are currently lacking. The most frequent form of degenerative dementia is Alzheimer’s disease [6], followed by Lewy body dementia, whose prevalence ranges up to 30.5% of all dementia cases [7,8]. Frontotemporal dementia is the third most widespread form, but it is acknowledged to be the first cause of early onset dementia [9,10].

In this scenario, any risk factor that can be modified must be taken into account in order to minimize the impact of progressive cognitive deterioration.

Interest in the role of homocysteine (HCy) began to rise in the 1990s [11]. Evidence for a potential role of such an amino acid in brain disease came from three data sources: (i) biochemical work on experimental animals, cell cultures, and humans, which clarified the mechanisms by which hyperhomocysteinemia (HHCy) can cause vascular and neuronal damage [12,13,14,15,16]; (ii) work on the role of HHCy in brain-damaged populations (e.g., stroke, dementia, etc.) [17,18]; and (iii) work on the effects of therapies with B vitamins and folic acid on some parameters, such as dementia severity, cognitive deficits, rate of progression of cognitive disorders, etc. [19,20,21].

The role of HCy in the genesis of cerebrovascular diseases (CVD) and dementias has long been debated due to controversial findings [11,20,22,23,24,25,26,27,28,29]. Nevertheless, a large corpus of empirical evidence now supports the view that HHCy indeed has a role in the genesis of those diseases, a notion that has also been supported by critical review of the findings that seemed to contradict it [12,30].

To date, HHCy is considered to be a risk factor for stroke, cerebral vasculopathy, and vascular dementia. A number of papers suggest that HHCy might be implicated in the genesis of dementias [20,24,26,31,32]. A recent Consensus Statement [30], based on the Bradford Hill criteria, concluded that HHCy is a modifiable risk factor for development of cognitive decline and dementias. They showed that in many clinical studies, the relative risk of dementia in elderly people for moderately raised HCy (albeit still within the normal range) ranges from 1.15 to 2.5, and the Population Attributable risk ranges from 4.3% to 31%. The authors also reviewed intervention trials in elderly with cognitive impairment and found that HCy-lowering treatment with B vitamins markedly slowed the rate of general and regional brain atrophy and of cognitive decline.

There are still two obscure points that need to be addressed: the first concerns the impact of HHCy on *different* forms of degenerative dementias, which are characterized by different pathogenetic and pathophysiological substrates and by diverse clinical phenotypes; and the second concerns the effects of HHCy on *specific* cognitive functions and on the relative brain areas/circuitries. In other words: Are there brain circuitries and cognitive functions that are particularly susceptible to the effects of HHCy? Beyond the intrinsic scientific value of any such a notion, knowing whether the action of HHCy determines selective impairment of specific cognitive domains could help to delineate the clinical target in studies exploring the efficacy of vitamin implementation, and would also explain why some studies found a sizable cognitive benefit while others failed to do so: Did the latter studies test the “wrong” cognitive domains?

When we investigated these crucial aspects in a review of the literature, we met obstacles of two main types. The first is that, in several papers, the criteria for including patients as affected by “cognitive impairment” or “dementia” are not clearly specified, and information about the specific forms of dementia is missing [33]. The second is that information on specific cognitive domains, which is central to our purpose, is typically unavailable: most studies in the field just report screening tests that explore the general cognitive status of the patients (e.g., the well-known Mini Mental State Examination test, MMSE). These are very crude measures, which fail to provide fine detail on the patients’ across-domains cognitive profiles [33].

This review has two main purposes: The first is to focus on the state of the art of the association between hyperhomocysteinemia and degenerative dementias, in order to understand if such an association varies with dementia type. The second aim is to summarize the state of the art of the complex relationship between homocysteine levels and *specific* cognitive functions. The two purposes are closely related to each other, inasmuch as the different forms of dementia have very different phenotypic profiles with impairment of specific cognitive functions.

For these purposes, we reviewed papers published after 1989 and that are listed in two main databases (PubMed/Medline and Web of Science). We chose the 1990 time constraint because scientific interest in the role of HCy and HHCy began in the early 1990s [11]. We used the MeSH browser and considered the following keywords, alone or in combination: “dementia”, “homocysteine”, “hyperhomocysteinemia”, “cognitive functions”, “cognitive decline”, “cognitive impairment”, “MCI”, Alzheimer’s Disease”, “Dementia with Lewy Body”, “Frontotemporal Dementia”, “Non-Alzheimer Dementia”, “Degenerative dementias”. In the present review, we focus on the state of the art relative to degenerative dementias, also including Mild Cognitive Impairment (MCI) [34,35]. A detailed revision of the evidence that is currently available on the relationship between HHCy and cognitive functions is provided (Section 6), also referring to the effects of folic acid and B vitamin supplementation (Section 6.3). For the sake of completeness, the aspects relating to the pathogenesis and the role of HHCy in dementia and cerebrovascular diseases are also summarized (Section 3, Section 4 and Section 5); since these latter are not the main focus of the present work, the reader will be referred to the most up-to-date reviews. Section 7 summarizes the neuroanatomical findings; that is, the relationships between HHCy and detectable morphological differences in the brain. Section 8 contains the concluding remarks and perspectives.

## 2. Homocysteine: From Normality to Pathology

Homocysteine (HCy) is a sulfur-containing amino-acid that is generated during methionine metabolism and is produced physiologically in all cells [13,14,15]. HCy can be degraded via the remethylation pathway, allowing for the production of methionine (which is fundamental for methylation-process), or converted, via the trans-sulfuration pathway, into cysteine (Figure 1). Interesting reviews on the subject are available [12,14,15,32]. Normal levels of HCy range between 5 and 15 micromol/L and, in physiological conditions, plasma total (t) HCy levels are <15 mol/L, as reported by the most relevant studies [12,14,15,32]. In healthy subjects, the physiological levels of HCy primarily depend on the dietary intake of methionine [36], folate [37], and vitamin B12 [38]. It follows that dietary restrictions or alcohol abuse can lead to HHCy. Physical inactivity also contributes to increase in HCy [39,40]. Age can lead to an increase in HCy levels; mild and moderate HHCy are frequently found in older people, probably due, at least in part, to a low intake of B group vitamins [22]. Different pathological conditions, such as diabetes, hypertension, and renal insufficiency, can be associated with elevated plasma total HCy levels [41,42,43,44]. In rare cases HHCy can be due to genetic mutations in folate- and methionine-metabolizing enzymes, namely, in methylene tetrahydrofolate reductase (MTHFR) and cystathionine β-synthase (CBS) [45].

## 3. Homocysteine and Vascular Contribution to Cognitive Impairment and Dementia

HCy is an acknowledged risk factor for cerebrovascular disease and stroke. This notion is critical insofar as many patients suffer from mixed forms of dementia: the concomitant presence of cerebrovascular diseases can aggravate the cognitive impairments resulting from the degenerative pathology (Figure 2).

The reader interested in the role of HCy in the pathogenesis of atherosclerosis and stroke can refer to some interesting reviews: Hainsworth et al. (2016) [12], Moretti and Caruso (2019) [13], McCully (2016) [15], and Nieerad (2021) [14]. Briefly, there are multiple mechanisms through which HHCy determines an increase in brain vascular pathology, particularly in cerebral small-vessel disease [46,47,48,49]. HHCy acts at both the molecular and the cellular level, involving neurons and vascular endothelial cells. There is evidence that HHCy determines neurotoxicity, suppression of neural network activity, and peroxidation of lipids [12,13,14,15]. At the same time HHCy damages blood vessels acting on vascular smooth muscle cells, by enhancing cell proliferation, fibrosis, and collagenosis. HHCy acts on endothelial cells, inhibiting cell proliferation and endothelial nitric oxide synthase [12]. High levels of HCy could contribute to cognitive decline by increasing the risk of small-vessels disease in the brain, which, in turn, causes microinfarcts, lacunes, lacunar strokes, and leukoaraiosis [48,49,50].

## 4. Homocysteine and Neurodegeneration Pathways

HCy is an agonist of endogenous glutamate receptors, NMDA [51,52], and it likely has an excitatory effect on neurons [53,54,55,56]. A second mechanism of HCy’s action consists of the activation of group-I metabotropic glutamate receptors, in competition with GABA or other inhibitory neurotransmitters [57]. In both cases, the effect is an increase in calcium influx. A positive correlation between HHCy and Abeta 1–40 deposition in the brain of AD patients has been reported [58]; HCy is thought to induce or potentiate the intracellular and extracellular accumulation of Abeta 42 and its neurotoxicity [25,59,60,61,62,63,64]. Similar findings were obtained with mouse models of AD [65]. There is also evidence that HCy increases the toxicity of Abeta on the vascular smooth muscle cells of small brain arteries [66,67,68,69]. Another interesting mechanism relies on HCy’s capacity to upregulate presenilin genes, in particular, the one regulating presenilin 1 (PS1) by DNA hypomethylation. Since PS1 promotes the amyloid precursor protein (APP) synthesis [70,71], the upregulation of the PS1 gene by HHCy determines an increase in APP, and as a consequence, it promotes the amyloid cascade sequence.

HCy seems to be also implicated in the metabolism of the tau protein, which is considered to be a “key protein” in several neurodegenerative diseases leading to dementia, collectively known as “tauopathies” [72,73,74]. This class of neurodegenerative disorders is characterized by neuronal and/or glial inclusions composed of tau, the microtubule-binding protein [72]. From the clinical viewpoint, tauopathies can present with several phenotypes that include dementias (i.e., frontotemporal dementia spectrum disorders), or movement disorders, or both (i.e., corticobasal degeneration and progressive supranuclear palsy). The tau protein binds microtubules to stabilize the cell cytoskeleton and is normally present in the cytosol of neurons and glial cells of the central nervous system [72,75,76]. The tau protein exists in six isoforms, and tauopathies are classified by which tau isoform is dominant in the cytoplasmic inclusions; thus, they are called 3R, 4R, or 3R:4R tauopathies. Alzheimer’s disease is usually considered as a secondary or non-primary tauopathy. In AD, intracellular neurofibrillary tangles, composed of equal ratios of 3R and 4R tau, and Aβ extracellular plaques coexist [72]. The exact relationship between the amyloid beta plaques and the process of tau aggregation is still unclear.

It is well known that in pathological conditions of tauopathy, tau is hyperphosphorylated. Tau hyper-phosphorylation inhibits the congregation of microtubules, and their precipitation produces the deposition of the neurofibrillary tangles. The activity of the protein phosphatase methyltransferase 2A (PP2A), which acts as a dephosphorylating system for the tau protein, is regulated by protein phosphatase methyltransferase 1 (PPM1), whose methylation is SAM (S adenosylmethionine)-dependent [77,78,79,80,81].

Thus, the reduced methylation capacity increases the concentration of hyperphosphorylated-tau (P-TAU) [82], with consequent neurofibrillary depositions.

Another mechanism by which HHCy acts in neurodegenerative diseases is through the reduction in dopamine turnover in the striatum, which, in turn, causes acceleration of the death of dopaminergic cells [83]. HCy has been found to have allosteric antagonist activity against D2 receptors [84]; in the middle portion of the third loop of the D2 receptor, there is an ARG-rich domain, which has high affinity for HCy [84].

For details as to the mechanisms of HCy neurotoxicity in neurodegenerative diseases and dementia, see the following reviews: [13,14,16,60]. Figure 2 summarizes the vascular and degenerative pathways through which HCy is thought to cause cognitive deficits.

## 5. Homocysteine and Degenerative Brain Diseases

In this section, we will synthesize the available evidence on the role of HHCy in subjects enrolled in the prodromal phase of dementia (MCI) or with a specific degenerative dementia. We excluded studies in which the diagnostic criterion for “dementia” or “cognitive impairment” was not fully specified, or studies which failed to define the exact type of dementia that was diagnosed. In this way, we ascertained not to discuss effects related to other factors inducing cognitive disorders in the elderly (systemic diseases, drugs, social marginalization, etc.).

### 5.1. Homocysteine and MCI

The relationship between MCI and HCy levels is still controversial. Some studies reported increased HCy levels in MCI individuals [85,86,87,88,89], while others failed to do so [90,91,92,93,94,95]. These conflicting findings could perhaps be related to the various ways MCI is diagnosed. Another potential confounding factor is the degree of cognitive impairment in individuals with MCI. For a review on this topic, refer to Ansari et al. [96]. A recent meta-analysis [97] concluded that HHCy is not appreciably associated with risk of MCI, and attributed positive findings to publication bias and low statistical power.

Contrasting results on the potential role of HCy in MCI also concern studies on vitamin supplementation [96]. In Lehman et al. [98], 30 MCI patients with increased HCy serum levels were orally supplemented with a high dose of a vitamin B12–B6–folate combination for 270 days. Cerebrospinal fluid levels of tau protein (CSF-tau) and albumin ratio were measured before and after treatment. The albumin ratio was significantly reduced after treatment, a finding that likely indicated tightening of the blood–brain barrier. Smith (2010) [99] performed a single-center, randomized, double-blind controlled trial of high-dose folic acid and vitamins B6 and B12 in 271 MCI individuals aged > 70. A subset (*N* = 187) were MRI-scanned both at the beginning and at the end of the study. Participants were randomly assigned to two groups: one treated with folic acid, vitamin B12, and vitamin B6, and the other with a placebo, for 24 months. The main outcome measure was the change in the rate of atrophy of the whole brain assessed by serial volumetric MRI scans. A total of 168 participants (85 in the experimental group, 83 in the placebo group) completed the MRI experiment. The authors found that the treatment effect depended on baseline HCy levels: the rate of atrophy in participants with high HCy baseline levels (>0.13 mmol/L) showed a sizable decrease with treatment (a 53% difference between the experimental and the placebo groups). Higher atrophy rates were associated with lower final cognitive test scores. The authors concluded that the accelerated rate of brain atrophy in elderly with MCI can be slowed down by treatment with HCy-lowering B vitamins. Despite the fact that this trial was designed to detect changes in rate of brain atrophy, a strong association between the latter variable and cognition was observed. However, in contrast with Smith et al.’s (2010) results, a randomized placebo-controlled trial with B vitamins [100] performed on 279 MCI patients found that Vitamin B12 and folic acid supplementation failed to reduce cognitive decline in older participants.

In conclusion, while evidence for a link between HHCy and MCI has been reported, the available literature is controversial and further investigation is needed. Evidence for B vitamin-mediated reduction in HCy and consequent improvement in cognition is opaque, too, and further randomized and controlled clinical trials are necessary to tackle the issue.

### 5.2. Homocysteine and Azheimer’s Disease

Many studies have explored the relationship between HCy and AD, starting from Clarke et al.’s (1998) pioneering research [101]. These authors performed a case–control study of 164 patients with a clinical diagnosis of AD (in 76 of which the diagnosis was histologically confirmed) and 108 controls. They found that AD was associated with low blood levels of folate and vitamin B12, as well as with increased HCy levels. The stability of HCy levels over time and the lack of a significant relationship with AD symptom duration allowed the authors to exclude that the increase in HCy was caused by AD.

Further evidence for the association between HCy and AD comes from many epidemiological studies [25,85,88,93,102,103,104,105], with some exceptions [94,106,107]. The HCy level is also associated with dementia severity [85], and HHCy is associated with accelerated cognitive decline in AD [108,109]. Some evidence from longitudinal studies is available, too. The most influential work is the Framingham Study performed by Seshadri and colleagues [110]. This involved a prospective investigation on 1092 dementia-free subjects, in which HHCy was found to be a strong, independent risk factor for dementia (including AD). The researchers found a graded increase in risk of dementia with increasing plasma concentration of HCy, after multivariate control for other putative risk factors for AD.

In contrast with the Framingham Study, in 2004, the Washington Heights–Inwood Columbia Aging Project (WHICAP) failed to find an association [107]. They randomly chose 909 elderly subjects from a cohort of Medicare recipients, and obtained longitudinal data from 679 subjects without dementia at baseline. The longitudinal analyses showed that the adjusted hazard ratio of AD for the highest quartile of HCy level was 1.4 (95% CI = [0.8, 2.4]); high HCy levels were not associated with a decline in memory scores over time. They concluded that high HCy levels were not associated with AD and were not related to a decrease in memory scores over time. However, some criticisms have been raised about this work: (i) insufficient statistical power; (ii) the generally high, and homogeneous, HCy concentrations levels of the recruited sample, as low variability swamps statistical associations, which can thus become undetectable; and (iii) methodological issues related to the long time elapsed between blood sample collection and processing, which might have affected tHCy measurements [111].

Ravaglia et al. (2005) [112] investigated the relationship between high plasma tHCy concentrations and AD risk in a sample of 816 elderly individuals without dementia in a longitudinal design. They found that over a follow-up of (on the average) four years, 112 subjects developed dementia, with 70 AD cases. Among the subjects with HHCy, the hazard ratio for dementia was 2.08 (95% CI = [1.31, 3.30]), and the one for AD was 2.11 (95% CI = [1.19, 3.76]). Low folate concentration was also associated with increased risk of both dementia (1.87; 95% CI = [1.21, 2.89]) and AD (1.98; 95% CI = [1.15, 3.40]). The authors concluded that HHCy and low serum folate are independent predictors of the development of dementia and AD [112].

Several meta-analyses concerning the HCy–AD relationship have been published. A recent one addressed modifiable factors associated with cognition and dementia [113], and involved 31 studies on incident AD; this revealed that high HCy (relative risk = 1.93; 95% CI = [1.50, 2.49]), low education, and low levels of physical activity were particularly strong predictors of incident AD. Similar results were obtained in another meta-analysis where AD and vascular dementia (VaD) were compared (Wang 2014) [79]: the HCy level in the AD group was slightly higher than that in the VaD group (mean difference = −4.76, 95% CI = [−7.59, −1.93]). A third meta-analysis [97] provided further evidence that a higher concentration of blood HCy is associated with a higher risk of AD. The most recent meta-analysis that we are aware of included prospective studies, and confirmed the association between high HCy and increased AD risk in the elderly [114].

### 5.3. Homocysteine and Non-Alzheimer Dementias

The number of studies addressing other, non-AD forms of degenerative dementia is quite low. We comment on them briefly below.

Zhang et al. [115] recently performed a retrospective case–control study including 132 dementia with Lewy body (LBD) patients, 264 AD patients, and 295 age-matched healthy controls. They found that elevated plasma tHCy levels were associated with LBD, and more strongly so than with AD. The authors also explored the relationship between tHCy levels and the duration of three types of symptoms, which involved the following tests: MMSE—a measure of cognitive functioning; CDR (Clinical Dementia Rating)—a measure of dementia severity, and the 12-item Neuropsychiatric Inventory (NPI)—a measure of neuropsychiatric impairment. Data analysis revealed that there was no significant relationship between the mean plasma tHCy concentration and the duration of symptoms, and the authors concluded that the observed associations should not be assumed to be a consequence of LBD.

In another study [116], the HCy levels were measured in nine FTD and nine LBD, and no significant different was found, in either group, with respect to healthy controls.

In some studies, non-Alzheimer dementia cases were collected and used as a control disease group [110,117,118]. However, these papers did not compare diagnostically defined subgroups—information as to the composition of such samples was scanty, and barely referring to the presence of vascular dementia cases and of other forms of degenerative dementias. In the Seshadri study [110], HHCy (plasma homocysteine > 14 μmol per liter) was associated with an increased risk of dementia of any type (relative risk = 1.9; 95% CI = [1.3, 2.8]) and of Alzheimer’s disease specifically (relative risk = 1.9; 95% CI = [1.2, 3.0]). In the Raszewski’s study [117], the MMSE score significantly correlated with HCy level only in the non-Alzheimer group.

## 6. Homocysteine and Cognitive Functions

### 6.1. Studies on Patients with Dementia

Previous review work [33] addressed the specific topic of the relationship between HCy and cognitive functions. The authors noticed how in the literature available at that time (thus, before 2000) the differential effect by HCy on separate domains of cognitive functions was typically neglected, with the exception of Riggs et al. [119], who reported that 70 healthy male volunteers, administered with memory and visuospatial tests, showed an association between high concentrations of HCy and poor spatial copying skills. Work published later than 2000, which we review below, kept neglecting the issue. Most of the studies exploring the role of HCy levels on the dementias do not report data from multi-domain neuropsychological examination [85,86,88,89,92,93,95,98,99,102,103,104,105,109,112,115,116,117,118,120,121,122,123,124,125,126,127,128,129,130,131,132,133,134,135,136,137,138,139,140,141,142,143,144]. Typically, authors focus on the general cognitive status of the patient, and thus report data from neuropsychological screening tools, such as the Mini Mental State Examination (MMSE). Several papers report that the diagnosis of dementia was formulated, also relying on data from neuropsychological batteries; however, data from such batteries are either not reported at all, or no analysis is reported that correlates specific neuropsychological tests with HCy levels. For instance, Lauriola et al. [85] used data from a neuropsychological battery to stratify the sample in four groups of increasingly severe cognitive impairment (MCI, mild, moderate, and severe dementia), and also grouped the patients in terms of dementia type (probable AD, possible AD, and vascular dementia). They found that the serum HCy concentration was higher in moderate and severe dementia patients than in MCI and mild dementia patients. Furthermore, VaD and possible-AD patients had higher serum HCy concentrations than probable-AD patients. However, no correlation was found between the HCy levels and performance in specific neuropsychological tests.

Nevertheless, there are a few exceptions to the rule. Data relative to the studies where specific cognitive functions were actually examined, in detail, are listed in Table 1. We distinguished between studies on dementia patients and studies on healthy individuals, and we excluded the studies exploring the effects of folate and B vitamins administration on cognition (however, given the strong link between HCy and folate/B-vitamins, some brief comments on those studies are provided).

To our knowledge, only four HCy-related papers have been published with an extensive neuropsychological evaluation of patients with dementia.

Sala et al. [94] studied the main cognitive domains of patients with subjective cognitive decline, MCI, AD, and VaD. They found a significant relationship with HCy values exclusively for two visuospatial tests: geometric figure copy (“easy” and “hard”) and the clock drawing test.

Faux et al. [145] tested healthy subjects and patients with MCI and AD. They found that composite z-scores of short- and long-term episodic memory, total episodic memory, and global cognition all showed significant negative correlations with HCy, in all three groups.

Our own lab investigated the relationship between HCy and the cognitive profile of 323 AD patients [146]; HHCy was associated with poor performance on memory tasks as well as on Luria’s motor planning test.

Moustafa et al. [147] tested 49 MCI patients. The authors focused on two specific cognitive functions: learning and generalization of rules. The anatomical regions usually associated with generalization of rules and learning are, respectively, the hippocampus and basal ganglia. The authors found that while the total HCy levels were higher in MCI individuals than in healthy controls, no difference between groups could be detected on the cognitive tests (learning and generalization tasks). In further analyses, MCI subjects were split into two subgroups, depending on their Global Deterioration Scale (GDS) scores: the first subgroup included individuals with very mild cognitive decline (vMCD, GDS = 2), and the second included individuals with mild cognitive decline (MCD, GDS = 3). They found that patients with MCD made more generalization errors than healthy controls and individuals with vMCD. They also found that total HCy levels correlated positively with generalization errors, but not with learning errors.

### 6.2. Studies on Healthy Subjects

Much more work exploring the association of HCy and specific cognitive functions has been performed on healthy subjects, in particular in elderly individuals [107,148,149,150,151,152,153,154,155,156,157,158,159,160,161,162,163,164,165,166,167,168,169]. All the studies we reviewed, showed that HCy plays an important role in cognitive processes, with the only exception that of Luchsinger et al. [107], who failed to find an association between HCy and memory. Nonetheless, a significant source of heterogeneity among the studies is that the cognitive domains were found to depend on the presence of HHCy. In most of them, the function showing statistical association with increased HCy in healthy individuals was episodic memory [148,150,151,156,157,158,160,161,162,163,166,168,169]. In other papers, higher HCy levels were found to be associated with a slow information-processing speed [150,154,165,169]. Other studies showed that higher HCy levels correlate with low performance in several tests tapping attention [149,156,158,161]. Still, other studies found a correlation between HHCy and executive functions [150,158,159,162,166]. In a smaller number of papers, the culprits were visuospatial functions [153,165], cognitive flexibility [158], working memory [157,167], language [159,162,166], fluid intelligence [154], logical abilities [167], and abstract reasoning [162]. Two papers reported an association between HHCy and all the cognitive domains that were explored [152,164]. There are also studies in which the relationship between HCy and specific cognitive functions—the feature we wish to address in the present review—cannot be extracted because cognitive results are presented in the form of latent variables extracted by means of factor analysis [159].

Clearly, there is large variability, both in the number of cognitive domains explored and in the types of neuropsychological tests used.

### 6.3. Folic Acid and B Vitamin Supplementation

Indirect evidence of the association between HHCy and specific cognitive functions comes from trials investigating the effect of folic acid and B vitamins supplementation (see also Section 7 below). In the FACIT study, a randomized, double-blind, controlled trial [170], the authors selected older adults according to their HCy levels; that is, not on the basis of their cognitive status. Five neuropsychological tests were used: word learning, concept shifting, Stroop, verbal fluency, and letter-digit substitution. They found that 3-year folic acid supplementation improved performance on tests that measure information-processing speed and memory. Other trials, exploring the effect of folic acid-containing supplements, studied the potential relationship with cognitive functions (see Durga et al. [170] for a review). Although the recruited samples were composed of individuals with unspecified or non-formalized cognitive complaints, or with a generic diagnosis of “dementia”, in these trials, too, evidence emerged that some cognitive domains may be more “sensitive” than others to folic acid or B12 supplementation.

Some studies explored the relationship between folate/B12 and cognition in healthy individuals (e.g., [171]), with results that the authors judged as inconclusive, because an association was found only in some cognitive domains (letter search, word recall, and verbal fluency) and not in others.

### 6.4. Studies on Animals

A broad literature investigated cognitive alterations in experimental animals (see Nieerard et al. [14] for a detailed review on this topic). Interestingly, the domains that were found to be most frequently impaired in animals with HHCy were spatial and recognition memory (80–90%). Working memory or psychomotor abilities were affected in approximately half of the relevant trials [14].

## 7. Homocysteine and Neuroanatomical Findings

A brief summary of the evidence available on the anatomo-functional correlates of HHCy can help understanding the mechanisms by which HCy associates with cognitive impairment. Animal studies revealed an association between cerebral atrophy and HHCy, with atrophy particularly involving the hippocampus (see Nieerard et al. [14] for an extensive review). Similarly, a number of cross-sectional studies on humans reported that regional brain atrophy is related to tHCy [127,133,172,173,174,175]. Some studies found that subjects with higher HCy levels show atrophy in several cortical areas (frontal, parietal, and temporal cortex) and related white matter [127,173]. Several studies found that HHCy is related to hippocampal atrophy [101,172,173,176].

Indirect evidence supporting the notion that HHCy contributes to degeneration of specific brain areas comes from studies investigating the role of B vitamin administration, such as the VITACOG trial [99,177]. This study revealed that in patients with MCI treated with B vitamin, the rate of global brain atrophy slowed down; further analyses focused on regional brain atrophy, and revealed that only some brain areas were significantly less atrophic in the treated than in the placebo group: the medial temporal lobe, the precuneus, the angular and supramarginal gyri [177]. These areas are the ones usually involved in AD.

In the same vein, the well-known Nun Study [178] deserves mentioning. The study aimed at determining whether serum folate was inversely associated with the severity of atrophy of the neocortex of 30 nuns. In 15 of them—those with significant AD lesions in the neocortex—the authors found a strong correlation between folate and atrophy (*r* = −0.80). Atrophy was reputed to be specifically related to low folate levels, because none of the other 18 nutrients, lipoproteins, or nutritional markers that were measured from the blood samples showed significant negative correlations with atrophy. While this study is clearly underpowered, and the correlation is thus likely overestimated [179], results are intriguing, and deserve further investigation.

In conclusion, evidence in favor of a link between HCy and brain atrophy is generally strong.

## 8. Conclusions and Suggestions for Future Research

The present review focused on two main issues concerning the study of the relationship between homocysteine (HCy) and neurological diseases. The first issue concerns the scarce amount of evidence from non-Alzheimer degenerative dementias. The second issue concerns the potential role that hyperhomocysteinemia (HHCy) has on specific cognitive functions.

Most research on HCy and degenerative dementias focused on Alzheimer’s disease (AD), with a large consensus on a strong association between HHCy and AD. While the focus on AD is not surprising, inasmuch as it is by far the most widespread type of dementia, the comparatively little attention to other forms of dementia is a significant drawback, especially because understanding whether HHCy is *specific* of AD, or rather, associated to dementia in general, would help elucidate the causal flow of such associations.

In the scanty literature on non-Alzheimer dementia, some work on MCI has been published in recent years, with the results being equivocal. As to dementia with Lewy bodies, the second degenerative disease after AD in epidemiological terms, we found only one study with an adequate sample size [115]. We could not find sizable studies on frontotemporal dementia (FTD): Hoffmann’s study [116] involved only nine patients with FTD and therefore has very low statistical power. We are not aware of any study dealing with the least frequent forms of non-Alzheimer dementia (corticobasal degeneration, progressive supranuclear palsy, etc.).

With regard to the association between HCy and specific cognitive functions, so far only a few studies analyzed the detailed cognitive profiles of subjects suffering from dementia. More evidence is available from samples of healthy subjects. Results are very inconsistent and variable, with different cognitive domains being identified as associated with HCy levels. The domain most frequently reported to be affected when HCy levels are high is episodic memory. Such evidence is in line with data coming from both animal research and neuroimaging data on humans, which show that the hippocampus—the most important brain region involved in episodic memory—is frequently found to be atrophic in individuals with HHCy.

In our view, the large variability and inconsistency of results across studies is unsurprising, given the equally large heterogeneity in the collected samples, in the types and ranges of cognitive tests used, and in the statistical techniques applied for data analysis. All these limitations should be addressed in future research, which should be more focused on investigating non-Alzheimer dementias and on a precise—*standardized*—neuropsychological characterization of the recruited patients.

Acquiring data on the differential impact of HHCy on separate cognitive domains could have not only theoretical but also practical value. So far, the main trend has been that of only using screening tests, without collecting, or analyzing, data from detailed cognitive evaluations, a practice that has been criticized by several authors [12]. For instance, the low sensitivity of screening tests has been considered as a key limitation in clinical trials testing the effects of B vitamin administration on cognition [12,180]. In this vein, identifying which cognitive domains are most compromised by HHCy would be of fundamental importance in order to provide a reliable outcome index in pharmacological trials.

The possibility that the progression of some forms of dementia might be slowed down by lowering the concentration of homocysteine would increase the hope of being able to maintain self-sufficiency into old age and would also be of considerable interest for public health purposes within primary dementia prevention campaigns. In the present scenario of a set of diseases whose progression cannot be stopped, being able to target specific modifications of manipulable risk factors would represent not only a valuable scientific aim but also a critical health policy goal.

It is worth mentioning that some recent preclinical work found that inhibition or inactivation of specific enzymes, such as cystathionine-β-synthase (CBS), exerts beneficial effects [181,182]. Progress in this field is expected to stimulate future studies, which might identify specific compounds to be used in the molecular therapy of homocysteine-related diseases, with clinically significant results.

## Figures and Tables

**Figure 1 biomedicines-10-02741-f001:**
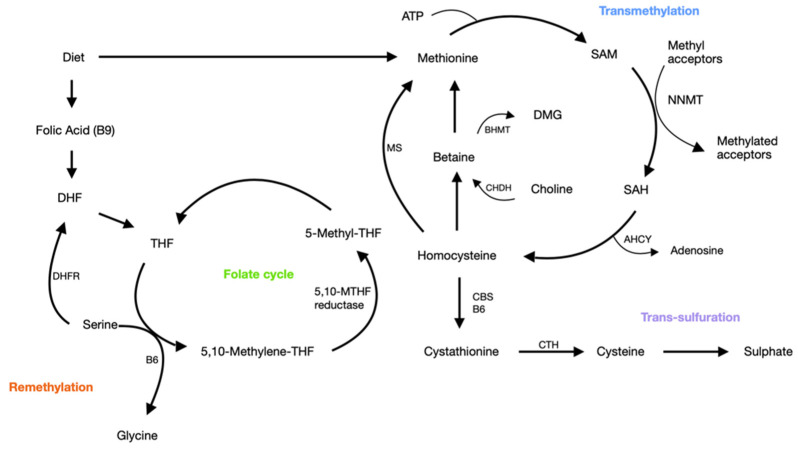
Pathways of homocysteine metabolism. AHCY, adenosylhomocysteinase; ATP, adenosine triphosphate; BHMT, betaine-homocysteine methyltransferase; CBS, cystathionine b-synthase; CHDH, choline dehydrogenase; CTH, cystathionine γ-lyase; DHF, dihydrofolate; DHFR, dihydrofolate reductase; DMG, dimethylglycine; MS, methionine synthase; NNMT, nicotinamide N-methyltransferase; SAH, S-adenosyl homocysteine; SAM, S-adenosyl methionine; THF, tetrahydrofolate; 5-MTHF, 5-methyltetrahydrofolate.

**Figure 2 biomedicines-10-02741-f002:**
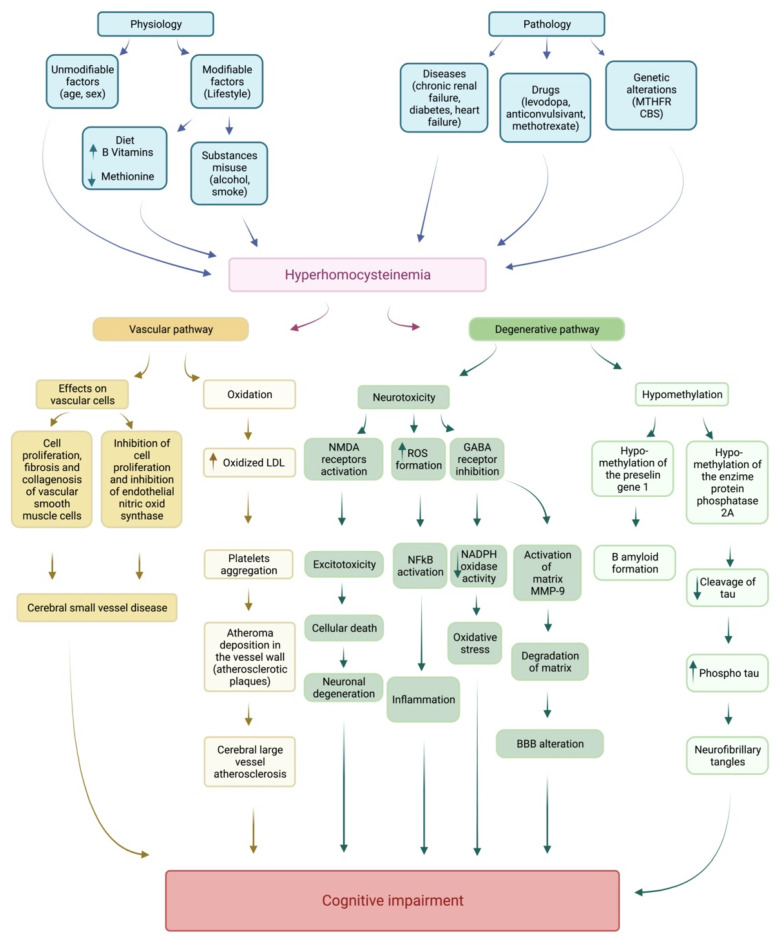
Homocysteine and neurodegeneration pathways. BBB, blood–brain barrier; CBS, cystathionine-β-synthase; GABA, gamma-aminobutyric acid; LDL, low-density lipoprotein; MMP-9, metalloproteinase-9; MTHFR, methylenetetrahydrofolate reductase; NADPH, nicotinamide adenine dinucleotide phosphate; NF-kB, nuclear factor kappa-light-chain-enhancer of activated B cells; NMDA, N-methyl-D-aspartate; ROS, reactive oxygen species.

**Table 1 biomedicines-10-02741-t001:** Studies where specific cognitive functions are examined.

Study	Study Design	Sample Size	Disease	Cognitive Domains	Neuropsychological Tests	Results (Association between HCy Levels and Cognitive Functions)
** *Subjects with dementia* **						
[94]	CS	325	27 SMC, 142 MCI, 139 AD, 17 VaD	Memory, language, executive functions, attention, and visuoconstructive functions	MMSE, Interview for Deterioration in Daily Living in Dementia, CDR, GDS, BDRS, CDT, WAIS (logical memory), FCRST, BVRT, CERAD (word list), BNT, Category-specific names test, SCWT, RPM, TMT A and B, Geometric figure copy	Significant relationship between HCy and performance in visuoconstructive functions
[145]	CS	1112	768 HC, 133 MCI, 211 AD	Short-term memory, long-term memory and global cognitive status	Four neuropsychological composite z-scores were computed: 1. Short-term episodic memory: logical Memory 1–Story A, CVLT-II immediate recall, RCFT 3-minute delay. 2. Long-term episodic memory:logical memory II, CVLT-II delayed recall, and RCFT 30-minutes delay. 3. Total episodic memory: all the tests from both sets 1 and 2 above. 4. Global cognitive: SCWT incongruent trials, category fluency switching, digit symbol coding, and the tests used for the Total episodic memory composite z-score.	Composite z-scores of short- and long-term episodic memory, total episodic memory, and global cognition all showed significant negative correlations with HCy, in all clinical categories
[146]	CS	323	AD	Memory, language, visuoperception, visuospatial abilities, executive functions, attention, constructional praxis and ideomotor praxis	MMSE, CPM, Digit span, Corsi’s Blocks, RAVLT, RCFT B, SCWT, Luria’s Motor Planning Test, Overlapping Figures of L.G., AB, Verbal Fluency, Phonological fluency, Naming, Reading, Word comprehension	HHCy was associated with poor performance on memory tasks and on Luria’s Motor planning test
[147]	CS	52	26 MCI, 26 HC	Striatal and hippocampal functions	MMSE, CDR, GDS. Cognitive task: Learning-and-Generalization test	Total HCy levels correlated positively with generalization errors, but not with learning errors
** *Healthy subjects* **						
[107]	L	909 (longitudinal data on 679)	Elderly HC	Memory, visuospatial abilities and language	BNT, controlled word association test, category naming, complex ideational material subtest, Boston Diagnostic Aphasia Evaluation (sentence repetition); WAIS-R similarities subtest, selective reminding test, non verbal identities and oddities of the DRS, Rosen Drawing Test, BVRT	High HCy levels not related to decline in memory scores over time
[148]	CS	21	Middle-aged HC	Memory	Ö-VMPT, WMS,visual memory test	Increased HCy associated with lower memory score
[149]	L	180	Adult HC	Memory and attention	MMSE, CVLT, DRS, SCWT	HCy levels correlated with SCWT baseline and follow-up scores. Change rates of HCy and change rates of SCWT scores also correlated
[150]	L	182	Older HC with hypertension	Speed of cognition, attention, episodic memory, working memory and executive functions	MMSE, computerized assessment battery and executive function tests, TMT A and B, Verbal fluency (FAS), Category fluency (Animals)	Higher HCy associated with more severe cognitive decline in speed of cognition, episodic memory and executive function
[151]	L	2189	Elderly HC (community-dwelling)	Memory	Kendrick Object Learning Test	Subjects with memory deficit had higher HCy and lower folate at follow-up
[152]	CS	1140	Randomly selected adult HC	Language conceptual reasoning, simple motor and psychomotor speed, eye-hand coordination/manual dexterity, executive functions, memory and visuoconstruction/visuoperception	BNT, Category fluency, Letter fluency, CPM, FTT (dominant and nondominant), Simple reaction time, Purdue pegboard (dominant, non-dominant and both hands), Purdue pegboard assembly, SCWT, TMT A and B, RAVLT, Visual memory and learning, RCFT-delayed recall 36, DS paired associate learning, RCFT-copy	Elevated HCy levels associated with poorer scores in all tested cognitive domains
[153]	L	321	Elderly HC (men)	Memory, language and spatial functions	MMSE, backward digit span, Word-list memory test, verbal fluency, constructional praxis	HCy associated with decline in constructional praxis, (spatial copying)
[154]	L	1257 (longitudinal data on 1076)	Older HC	Memory, information processing speed, reasoning	MMSE, Auditory Verbal Learning Task, Coding Task, CPM	Higher HCy at baseline was negatively associated with prolonged lower cognitive functioning and a faster rate of decline in information processing speed and fluid intelligence
[155]	CS	400	HC (men, age 40–80)	Short-term memory, speed of information processing, long-term memory, word fluency, cognitive flexibility and verbal intelligence	MMSE, Digit span, DS, RAVLT, Doors test, Verbal fluency (letters N and A, animals, and occupations), TMT A and B, Dutch Adult Reading Test	HCy associated with processing capacity and speed
[156]	CS	639	HC	Memory, attention	Electronic Memory Span apparatus, electronic Attention Span apparatus, Functional Activities Questionnaire	Negative correlation between HCy and: attention span, delayed (but not immediate) memory recall
[157]	CS	200	HC (women, age 56–67)	Verbal and working memory, executive functions, intelligence	CVLT-II, Ten unrelated words, WAIS Letter-Number Sequencing, NART-R	HHCy associated with poor performance on combined score of verbal and working memory
[158]	CS	170	HC (post-menopausal women)	Memory, executive functions, psychomotor speed, reaction time, attention and cognitive plasticity	MoCA, verbal memory test, FTT, SDMT, SCWT, SAT, continuous performance test	HHCy correlated with increased risk of de- cline in executive functions, complex attention, cognitive flexibility, and memory
[159]	CS	228	HC (community-dwelling)	Memory, executive functions and language. Factor analysis was used to summarize them into two orthogonal factors: memory and executive/language factors	MMSE, CDR, immediate recall, delayed recall, recognition, savings, TMT A and B, diamond cancellation, letter cancellation, animal fluency, Shipley vocabulary, BNT	In very old, non demented community dwellers, high HCy levels associated with poor executive-language functioning but not with poor memory
[160]	CS	466	HC	Intelligence, psychomotor processing speed, verbal fluency, category fluency and memory	MMSE, NART-R, WAIS—III-digit symbol coding, Verbal fluency (FAS test and animal fluency), WMS	HCy consistently associated with poorer performance in tests assessing visual memory and verbal recall
[161]	CS	3914	HC, age > 65	Verbal fluency, attention, memory	MMSE, The Isaacs Set Test (colors, animals, fruits, cities), TMT A and B, BVRT, Five-Word Memory Test	HCy associated with Benton Visual Retention Test and Trail Making Test
[162]	CS	2096	HC (dementia- and stroke-free)	Abstract reasoning, memory, language, executive functions	WAIS (Similarities subtest, Paired-Associates Learning), Logical Memory (immediate recall; delayed recall; delayed recognition), Visual Reproductions (immediate recall; delayed recall; delayed recognition), Halstead-Reitan test (trails A and B), Hooper Visual Organization Test, BNT	Older adults showed negative association between HCy and several cognitive tasks: abstract reasoning; verbal and visual memory; concentration, scanning, tracking, executive performance; visual organization; object naming and language. No such effects were found in younger and middle-aged adults
[163]	L	144	HC	Cognitive speed attention and information processing memory	Letter-Digit Coding test, SCWT, Word Learning Test; Delayed Recall	HCy correlated negatively with cognitive performance on Word Learning Test at baseline. Baseline HCy correlated negatively with follow up scores (up to 6-years) on SCWT and Word Learning Test
[164]	CS	Baseline: 1241; 2-years FU: 1151; 3-years 986; 4-years 782	Middle-aged HC	Attention and psychomotor speed	MMSE, TMT B, DSST of the WAIS-R, FTT	Higher concentrations of HCy related with poorer performance on all neuropsychological tests
[165]	CS	451	Middle-aged HC	Attention, memory, executive function, information processing speed, visuospatial and constructional ability	MMSE, ADL, Digit Span, Spatial Span, RAVLT, Visual Reproduction, Categorical Verbal Fluency, Design Fluency, SDMT, TMT A and B, Block Design	HCy specifically associated with constructional ability and information processing speed, whereas folate was associated specifically with episodic memory and language ability
[166]	L	274	HC	Global cognition, episodic memory, executive functions, language, psychomotor speed	MMSE, immediate word recall test, SCWT, category fluency test, bimanual Purdue Pegboard test, letter-digit substitution test	High baseline HCy associated with poor episodic memory, executive functions and verbal expression
[167]	CS	334	HC from the Aberdeen 1921 (*N* = 186) and 1936 (*N* = 148) Birth Cohorts	Memory, intelligence, language, attention, executive functions and visuospatial functions	MMSE, NART, RPM, RAVLT, WAIS (DS and block design subtests)	In 1921 Cohort, HCy negatively correlated with RPM, DS, and block design scores
[168]	CS	2470	HC (age > 60)	Memory	MMSE, delayed story recall, delayed word recall	HHCy was related to poor memory recall
[169]	CS	1077	Elderly HC	Psychomotor speed and memory	MMSE, Abbreviated SCWT, Letter-Digit Substitution Task,Verbal fluency test, Paper-and-Pencil Memory Scanning Task, 15-word verbal learning test (based on RAVLT)	Increasing HCy levels associated with lower psychomotor speed and memory scores

Legend: AD = Alzheimer’s disease; CERAD = Consortium to Establish a Registry for Alzheimer’s Disease; CS = cross-sectional study; FU = follow-up; HC = healthy controls/subjects; HCy = homocysteine; HHCy = hyperhomocysteinemia; AB = Apraxia Battery; ADL = Activities of Daily Living; BDRS = Blessed Dementia Rating Scale; BNT = Boston naming test; BVRT = Benton Visual Retention Test; CDR = Clinical Dementia Rating Scale; CDT = Clock drawing test; CPM = Raven Coloured Progressive Matrices; CVLT/CVLT-II = California Verbal Learning Test; DRS = Mattis Dementia Rating Scale; DS = Digit Symbol; FCRST = Free and cued selective reminding test; FTT = Finger Tapping Test; GDS = Global Deterioration Scale; L = longitudinal study; MCI = Mild Cognitive Impairment; MMSE = Mini-Mental Status Examination; MoCA = Montreal Cognitive Assessment; NART-R = National Adult Reading Test—Revised; Ő-VMPT = Őktem—Verbal Memory Processes Test; RAVLT = Rey Auditory Verbal Learning Test; RCFT = Rey Complex Figure Test; RPM = Raven’s Progressive Matrices; SAT = Shifting Attention Test; SCWT = Stroop Color and Word Test; SDMT = Symbol Digit Modalities Test; SMC = Subjective Memory Complaint; TMT = Trail Making Test; WAIS (-R) = Wechsler Adult Intelligence Scale (Revised); WMS = Wechsler Memory Scale.

## Data Availability

Not applicable.

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
