# Peer review of "Homocysteine, Cognitive Functions, and Degenerative Dementias: State of the Art"

_biomedicines, 2022, doi:10.3390/biomedicines10112741_

Round 1
Reviewer 1 Report
Dear Editor
In the present review paper, the authors introduce “Homocysteine, cognitive functions and the degenerative dementia”.
In general, the topic of the paper is interesting and the paper is well written. But, there are some points to improve the article.
-The authors can explain the mechanism of homocysteine‐induced cognitive dysfunction, neurodegeneration, and etc.
-The paper needs proper punctuation and language editing.
-The authors should prepare at least some attractive figures for the manuscript (For example: 1. homocysteine and neurodegeneration pathways; 2. Clinical treatments).
- The important point is that, they can introduce novel application of molecular therapy using critical amino acids such as homocysteine in molecular medicine
Reviewer 2 Report
This review article on Homocysteine, cognitive functions and the degenerative dementias: state of art is aimed at to understand the actual role of homocysteine in the pathogenesis of cognitive decline and to improve the diagnostic accuracy, differential diagnosis and the prognostic implications.
The introduction, tables, concluding remarks and references are framed correctly. Overall, the article is informative for the scientific fraternity. I recommend the paper for publication after rectifying the below mentioned minor errors
The authors could have included the figure showing the Homocysteine metabolism pathway ie remethylation to methionine, which requires folate and vitamin B12 , which will be more informative.
Line No.136: insofar as (Formatting error)
Line 414: verbal fluency, letter-digit substitution (instead of coma and is required)
Line 469: , inasmuch as ( formatting error)
In the references quoted Page numbers are missing:
Ref. Nos. 14,31,45,58,65,73,89,91,93,98,101,102,103,104,107,110,112,125,129 &131
